# Bioconversion of Callus-Produced Precursors to Silymarin Derivatives in *Silybum marianum* Leaves for the Production of Bioactive Compounds

**DOI:** 10.3390/ijms22042149

**Published:** 2021-02-21

**Authors:** Dina Gad, Hamed El-Shora, Daniele Fraternale, Elisa Maricchiolo, Andrea Pompa, Karl-Josef Dietz

**Affiliations:** 1Botany and Microbiology Department, Faculty of Science, Menoufia University, Shebin EL-Koum 32511, Egypt; 2Biochemistry and Physiology of Plants, Faculty of Biology W5, Bielefeld University, 33501 Bielefeld, Germany; karl-josef.dietz@uni-bielefeld.de; 3Botany Department, Faculty of Science, Mansoura University, Mansoura 35511, Egypt; shora@mans.edu.eg; 4Department of Biomolecular Sciences, University of Urbino “Carlo Bo” Via Donato Bramante, 28, 61029 Urbino, Italy; daniele.fraternale@uniurb.it (D.F.); e.maricchiolo@campus.uniurb.it (E.M.)

**Keywords:** bioconversion, flavonolignans, ESI-MS, HPLC, leaf extract, silymarin, *Silybum marianum*

## Abstract

The present study aimed to investigate the enzymatic potential of *Silybum marianum* leaves to bioconvert phenolic acids produced in *S. marianum* callus into silymarin derivatives as chemopreventive agent. Here we demonstrate that despite the fact that leaves of *S. marianum* did not accumulate silymarin themselves, expanding leaves had the full capacity to convert di-caffeoylquinic acid to silymarin complex. This was proven by HPLC separations coupled with electrospray ionization mass spectrometry (ESI-MS) analysis. Soaking the leaf discs with *S. marianum* callus extract for different times revealed that silymarin derivatives had been formed at high yield after 16 h. Bioconverted products displayed the same retention time and the same mass spectra (MS or MS/MS) as standard silymarin. Bioconversion was achieved only when using leaves of a specific age, as both very young and old leaves failed to produce silymarin from callus extract. Only medium leaves had the metabolic capacity to convert callus components into silymarin. The results revealed higher activities of enzymes of the phenylpropanoid pathway in medium leaves than in young and old leaves. It is concluded that cotyledon-derived callus efficiently produces compounds that can be bio-converted to flavonolignans in leaves tissue of *S. marianum*.

## 1. Introduction

The production of chemical substances in plants with pharmaceutical activities dates back to ancient times, and since then many bioactive molecules with pharmacological property have been produced in plants for different uses [1]. The production of molecules for pharmaceutical use can be spontaneous or engineered either with recombinant DNA biotechnology procedures [2] or through technological approaches that use more conventional procedures [3].

*Silybum marianum* (L.) Gaertn (milk thistle, family Asteraceae) is an annual or biannual winter medicinal plant native to the Mediterranean region [4]. Active components derived from the plant play a significant role in pharmacology and their specific medicinal and pharmaceutical applications. Silymarin is the most bioactive compound in *S. marianum,* which is a mixture of flavonolignans including silybin (SB), isosilybin (ISB), silydianin (SD), silychristin (SC), and taxifolin (TXF). It can be obtained from the fruits of the milk thistle [5,6,7]. The whole plant is used for various medicinal purposes, and fruit extracts are administered to treat liver diseases including cirrhosis [8].

Silymarin is an extract from the fruits of *S. marianum* containing flavonolignans and fractions of polyphenolic polymers. Flavonolignans are the main constituents that develop in the fruit pericarp of *S. marianum* [9]. Flavonolignans are a flavonoid subgroup characterized by an additional monolignol moiety [10,11]. Many studies describe the chemistry and pharmacological activities of flavonolignans [12,13,14]. The main medical functions of silymarin relate to its antioxidant activity by reducing the formation or decomposing free radicals and lipid peroxides. Moreover, silymarin is used to treat liver disease caused by alcohol abuse, viral hepatitis and liver disease caused by toxins [15,16]. Additionally, silymarin exhibited cytoprotective effects on human prostate and breast cells exposed to carcinogenic agents [17]. Moreover, the flavonolignan silybin displays chemopreventive and chemosensitizing activity against various cancers [18]. Thus, drugs containing flavonolignans are utilized worldwide.

Flavonoids are important secondary metabolites with multiple functions in stress defense, e.g., by repelling insects and herbivores, fighting pathogens, mitigating damage by ultraviolet light, but also in beneficial biotic interactions by attracting insects for pollination purposes [19,20,21]. The pathway of flavonoid biosynthesis starts with chalcone synthesis by chalcone synthase (CHS). CHS catalyzes the addition of a cinnamoyl-CoA unit to three malonyl-CoA units and subsequent cyclization. Chalcones function as precursors for a wide range of flavonoid derivatives found in plants. Phenylalanine ammonia lyase (PAL) and CHS are the committed steps of plant general phenylpropanoid metabolism and of flavonoid biosynthesis [22,23]. Flavonoids have antiviral, anti-inflammatory, and antioxidant activities associated with their potential to scavenge free radicals and also to modulate pathways of cell signalling [24,25].

Few studies have been carried out to investigate the synthetic pathway and the genes involved in flavonolignans synthesis in plants [26,27]. However, flavonolignans synthesis and regulatory mechanisms are still difficult to understand [28]. The main active molecules of *S. marianum* fruit extract belong to the flavonolignans and are called silybins. The importance of silybins has been confirmed in several studies. The compound is composed of two main units, taxifolin and a phenylpropanoid unit, which in this case is coniferyl alcohol. Both units are joined together by a dioxan ring to form silybin [29]. Silibinin is a mixture of silybin A and silybin B. Silybin is the biologically most active component and contributes to 50–70% of the silymarin activity [30].

Experimental studies of silybin synthesis in vitro have shown that silybin is produced by oxidative coupling of coniferyl alcohol and taxifolin [31,32]. Although silybin is the best-studied flavonolignan and was detected first [33,34], only one full-length and four partial cDNA fragments have been cloned from milk thistle [26,27]. It is difficult to clarify the biosynthetic and regulatory mechanisms of silybin biosynthesis, because of the limitation of genetic information on *S. marianum*. Studies in other plant systems make use of secondary metabolite accumulation caused by an elicitor or other inducer. The kinetics of expressional induction eases the identification of involved genes. In other cases, secondary metabolites displayed a high degree of turnover in species of interest where breakdown was investigated [35].

The aim of the present work was to investigate the capacity of leaf tissue to synthesize silymarin from phenolic precursors accumulating in *S. marianum* callus. It should be noted that cotyledon-derived callus was unable to synthesize silymarin [36]. Leaves lack the capacity to accumulate silymarin compounds. We established a novel bioconversion system for silymarin synthesis by combining callus extract and leaf tissue of specific developmental state. In line with the changing conversion capacity, activities of involved enzymes peaked in a particular developmental stage. These enzymes included PAL, CHS, shikimate *o*-hydroxycinnamoyl transferase (HCT), cinnamoyl-CoA reductase (CCR), cinnamyl-alcohol dehydrogenase (CAD), and caffeoyl-CoA-methyltransferase (COMT).

## 2. Results

### 2.1. Bioconversion of Callus-Produced Compounds to Silymarin Derivatives

#### 2.1.1. Leaf Discs for Bioconversion

The possibility of transforming products derived from *S. marianum* calli culture was explored using *S. marianum* leaves. To this end, discs prepared from mature leaves were soaked in callus extracts for various times (Figure 1). At indicated times, leaf discs were extracted and analyzed by HPLC. *S. marianum* callus derived from cotyledons produced chlorogenic acid (CGA) and dicaffeoylquinic acid (DCQ) but no silymarin (Figure 2a). Both compounds have been described by [36] in the same callus. It should be noted that the leaves did not contain either CGA, DCQ, or silymarin (Figure 2b). The DCQ-peak was strongly decreased after 4 h of incubation. Instead, a new peak at 19.9 min appeared (Figure 2c,d), which subsequently lost its intensity again after 8 h. After 16 h, a particular peak pattern appeared with retention times of the peak maxima at 19.1, 20.0, 25.1, 25.8, 27.0, and 27.4 min (Figure 2e). These five peaks appearing in Figure 2e were identical to those of standard silymarin (Figure 2g). Thus, DCQ proved to be a good substrate for silymarin biosynthesis by leaves of medium age. Also, degradation of silymarin was observed after 24 h (Figure 2f).

Analyses by ESI-MS and MS/MS in the negative ion mode identified the compound at peak 19.9 min (Figure 2c) as apigenin-7-*O*-glucuronide. The dominant mass for the intermediately synthesized unknown peak was 445 *m*/*z* and its fractionation mass was 268.8 *m*/*z* (Figure 3). The dominant mass for the produced silymarin derivatives after incubation for 16 h (Figure 4a–c) was 481.1 *m*/*z*, which was the same mass as found in the standard silymarin. The dominant product ions of peaks at 25.1, 25.8, and 27 min (Figure 2e) generated from the precursor ion 481.1 *m*/*z*. corresponded to the molecular mass of silybin A, silybin B, and isosilybin A&B based on their fragment masses (Figure 4a–c).

#### 2.1.2. Leaf Age Greatly Affects Silymarin Bioconversion

In order to understand if the age of the leaves had an effect on the bioconversion of the compounds derived from callus cultures, we tested the conversion using leaves of different age. Our results revealed that leaf age greatly affects silymarin bioconversion since young as well as old leaves lacked the ability for bioconverting DCQ to silymarin derivatives after 16 h of incubation, while the intermediate compound apigenin-7-*O*-glucuronide was still generated (Figure 5).

#### 2.1.3. Leaf Extract for Bioconversion

The leaf extract had the same potential as leaf discs for the conversion of callus compounds to silymarin. The buffer solution influenced the bioconversion of CGA and DCQ (Figure 6a) to silymarin since cold leaf extract with phosphate buffer at pH 7.2 expressed the ability to synthesize silymarin from *S. marianum* calli extract (Figure 6c) compared to standard silymarin (Figure 6d). On the other hand, cold leaf extract with citric acid buffer (pH 4.6) lacked the ability to produce silymarin derivatives (Figure 6b). Hot plant extract was not effective for bioconversion of callus compounds (Figure 7a) and did not induce silymarin biosynthesis (Figure 7b,c) neither in phosphate nor citrate buffer compared with standard silymarin (Figure 7d) demonstrating the involvement of heat-labile components in the conversion pathway.

The silymarin produced by the leaf disc or leaf extract from callus compounds was quantified by HPLC, and the result is presented in (Table 1). The leaf discs or leaf extract were incubated with callus extract for 16 h or 3 h, respectively, to produce silymarin. Extract of one gram of *S. marianum* dry callus produced 8.4 mg of silymarin by using leaf discs, while 4.6 mg of silymarin was produced with leaf extract. Leaf discs produced the highest value of silydianin and isosilybin A and B, namely 2.2 and 1.6 mg·g dry weight^−1^, respectively. The conversion assays with leaf extract produced silymarin derivatives at lower amounts ranging between 0.72 and 1.09 mg·g DW^−1^.

### 2.2. Enzymes Activities

The peculiar difference in conversion efficiency between leaves of different developmental state prompted us to determine enzyme activities of the phenylpropanoid pathway. The activities of relevant enzymes were estimated in *S. marianum* leaves spectrophotometrically and are presented in (Figure 8). The studied enzymes are involved in flavonolignans biosynthesis such as chalcone synthase (CHS), flavonoid 3′-monooxygenase (F3′H), shikimate *O*-hydroxycinnamoyl transferase (HCT), cinnamoyl-CoA reductase (CCR), cinnamyl-alcohol dehydrogenase (CAD), and caffeoyl-CoA *O*-methyltransferase (COMT). These enzymes were detected in different developmental stages of *S. marianum* leaves (young, medium, and old). The medium leaves recorded the highest activities of all tested enzymes. Enzyme activity in young leaves ranged between 30–50% of the medium leaves, while old leaves displayed the least activities of these enzymes with 14–30% of medium leaves.

## 3. Discussion

Leaf and leaf cell development is characterized by five phases, initiation, expansion, maturation, mature state and senescence. The metabolic and genetic programs overlap in these phases, all the more because different areas of a leaf adopt different states [37,38]. Medium leaves may be assigned to the transition from expansion to maturation, while young leaves were in the expansion phase, and old leaves in the late mature state. The metabolic competence of medium leaves to convert DCQ to the silymarin complex is a novel and important finding. The conversion capability and efficiency were revealed by HPLC separation and verified by mass spectrometry.

The main compounds produced by *S. marianum* callus are known as CGA and DCQ as shown by [36] (Figure 2a). Moreover, the leaves of *S. marianum* did not produce silymarin as shown in the chromatographic analysis (Figure 2b). It was reported that silymarin was undetectable in the mottled thistle leaves or stems and the content of silymarin between parts of the plant ranged from zero in stems and leaves to 14.7 g/kg in seeds [39]. The unknown peak eluting at 19.9 min (Figure 2c) was identifiable as apigenin-7-*O*-glucuronide by virtue of its fragments by MS and MS/MS. The parent and daughter masses correspond to apigenin-7-*O*-glucuronide according to [40]. Transient occurrence of apigenin-7-*O*-glucuronide, which is a major flavonoid of *S. marianum* [40], suggested that calli lack expression of enzymes required in early steps of silymarin synthesis. Biogenetically, apigenin is a product of the phenylpropanoid pathway and probably obtained from DCQ by CHS to form chalcone, which is further isomerized by chalcone isomerase to form naringenin. Finally, a flavanone synthase oxidizes naringenin to apigenin [41,42].

The dominant product ion of the peak at 25.1 min generated from the precursor ion of 481.1 *m*/*z* was 300.9 *m*/*z*. The obtained *m*/*z* values corresponded to the molecular mass of silybin A and B since both masses are in line with the data reported by [7]. They also matched the product ions of silybin A and B collected from standard silymarin (SIGMA). The assignment to silybin A occurred based on the HPLC elution profile. We could not identify valid diagnostic peaks to distinguish silybin A and B, which is quite similar to the data reported in [7]. Nevertheless, the MS and MS^2^ analysis together with the HPLC elution profile confirmed the principle assignment. The next dominating peak was eluted from the HPLC at 25.8 min and acquired an ESI-MS mass of 481.1 *m*/*z* and a main MS/MS fragment of 300.9 *m*/*z*. This corresponds to the molecular mass of silybin B. The last peak eluted at 27 min in the HPLC chromatogram was identified as isosilybin A and B based on its mass of 481.1 *m*/*z* and the fragment mass of 453 *m*/*z* in the MS/MS-mode [7].

Flavonolignans are synthesized by oxidative coupling of a flavonoid (taxifolin) and a coniferyl alcohol [43]. Previously it was shown that *S. marianum* peroxidases are able to catalyze the oxidative coupling of taxifolin and coniferyl alcohol to silybinins and the synthetic activity was mainly associated with the extracellular compartment [43].

Coniferyl alcohol and taxifolin are probably synthesized in vitro from DCQ (callus compounds) by caffeoyl-CoA pathway. The proposed pathway for silybin biosynthesis from DCQ (*S. marianum* callus components) starts from coniferyl alcohol and taxifolin as substrates for flavonolignane biosynthesis (Figure 9). Coniferyl alcohol and taxifolin are synthesized through the *p*-coumaroyl-CoA and caffeoyl-CoA pathways. This pathway involves a series of enzymes, namely PAL, shikimate *O*-hydroxycinnamoyl transferase (HCT), cinnamoyl-CoA reductase (CCR), caffeoyl-CoA *O*-methyltransferase (COMT), cinnamyl-alcohol dehydrogenase (CAD), flavonoid 3′-monooxygenase (F3′H), and CHS [44,45].

The first steps in plant flavonoid biosynthesis are catalyzed by several enzymes. The key metabolic roles make these enzymes a desirable target for tracking regulatory mechanisms and engineering the pathways. Therefore, *S. marianum* leaves were used in the present study as a source for these enzymes that are required for converting callus compounds into silymarin derivatives.

PAL catalyzes the conversion of l-phenylalanine to ammonia and trans-cinnamic acid. PAL is the first and committed step in the phenylpropanoid pathway and is therefore involved in the biosynthesis of phenylpropanoids, flavonoids, and lignin in plants [46]. CHS plays a key role in the biosynthesis of flavonolignan in different parts of *S. marianum* [26,27,28,29,30,31,32,33,34,35,36,37,38,39,40,41,42,43,44,45,46,47]. CHS of *S. marianum* expressed the highest activity in medium leaves compared to young and old leaves. The activity of CHS has been shown to increase during early stages in the development of oat leaves followed by a marked decrease in older leaves [48]. This may reveal a reasonable explanation for the negative effect of old leaves for silymarin biosynthesis. The increase in silybin content has been accompanied by up-regulation of the CHS1, CHS2, and CHS3 gene expression, which are involved in the silybin biosynthetic pathway [49].

Besides being part of the plant developmental program, CHS gene expression is induced in plants under stress conditions [47,48,49,50]. F3H or flavonoid 3-hydroxylase catalyzes the conversion of flavonoid to 3′-hydroxyflavonoid in presence of NADPH [51]. A flavanone-3-hydroxylase (F3H) was identified that catalyzed the conversion of eriodictyol to taxifolin [52], hence the oxidative coupling of coniferylalcohol and taxifolin serves as the last but key node in silybin biosynthesis [53].

Silymarin degradation occurred after 24 h soaking (Figure 2f), which is in agreement with the findings of [43] who reported that silymarin compounds are also degraded by suspension culture peroxidases. It has been suggested that high peroxidase activity found in several plant cell cultures may contribute to the very low amounts of secondary products found in in vitro cultured cells [54,55,56]. On the other hand, the increase of peroxidase activity after elicitation is a well-documented phenomenon, and a number of roles during these processes have been proposed.

Generally, enzymes of phenylpropanoid pathway are known for their pH-dependent activity. Therefore, the impact of pH changes with phosphate buffer of pH 7.2 or citric acid buffer of pH 4.6 on silymarin bioconversion was determined. Interestingly, leaf extract of *S. marianum* with phosphate buffer solution at pH 7.2 had the ability to produce silymain flavonolignan. Previous work has reported that the highest activity of CHS was at nearly pH 8.0 and this is consistent with the pH optimum of known CHIs (pH 6.6–8.2) [57,58,59].

## 4. Materials and Methods

### 4.1. S. marianum Callus Production

*S. marianum* callus was generated from cotyledons induced on solid MS media supplemented with 0.5 mg/L of 2,4-D, 25 mg/L Asn, 0.01mg/l BAP, and 25 mg/L inositol (media I) [36]. Incubation of all cultures occurred in an incubation chamber at 25 ± 2 °C under dark condition. Callus was transferred three times to new media every 21st day and harvested after 63 days.

### 4.2. Callus Eextract Preparation

The dried powder of *S. marianum* callus was defatted with petroleum ether. The callus components were extracted by soaking the tissue powder in methanol (85% *v*/*v*) for 24 h with shaking. The homogenate was centrifuged at 3000 rpm for 10 min. The supernatant was decanted in a glass vial and evaporated to dryness. The crude callus extract was dissolved in 85% ethanol.

### 4.3. Metabolite Profiling in S. marianum Leaves

Leaves of greenhouse-grown *S. marianum* plants were washed several times with distilled water and leaf discs of 5 mm were prepared. Crude methanolic extract of 100 mg dry weight of *S. marianum* callus derived from cotyledons was dissolved in 85% ethanol. About 15 discs of *S. marianum* leaves were immersed in a reaction mixture composed of 250 μL of 1 mM CaCl_2_, 200 μL distilled water, and 150 μL of the previously prepared callus extract. The reaction mixture was incubated at room temperature for 4, 8, 16, and 24 h. Leaf discs were collected and washed with distilled water.

Leaves were extracted and metabolites measured using HPLC. Leaf discs were grinded with a pestle with small amounts of sterilized washed sand in sterile Eppendorf tube. Petroleum ether was added to the grinded leaves for defatting. The suspension was shaken and centrifuged at 2000 rpm for 5 min for sedimentation. The supernatant was discarded, methanol (1 mL) was added to the sediment, and the suspension was stirred several times. The homogenate was centrifuged at 2000 rpm for 5 min. The supernatant was filtered using sterilized 0.2 μm sterile micro-filter. The methanolic extract was transferred to glass vials and kept under hood for evaporation. Extracted silymarin was obtained as soft yellow powder. The dried residue was re-suspended in 100 µL ethanol and considered as ‘silymarin sample’ used for HPLC analysis after filtration. Control tests were carried out where discs were used without callus extracts. Leaves of different stages (young, medium, and old leaves) were used under the same conditions.

### 4.4. Leaf Extracts Prepared in Phosphate or Citric Acid Buffer for Bioconversion

Green leaves of *S. marianum* (100 mg) were extracted in 2 mL of 100 mM phosphate buffer (pH 7.2) or 100 mM citric acid buffer (pH 4.6). The homogenate was centrifuged at 2000 rpm for 10 min. The supernatant was kept in a glass vial and the residue was discarded. The reaction mixture contained 95 µL of the previously prepared supernatant, 100 µL of phosphate buffer, and 5 µL of callus extract. The reaction mixture was incubated at 37 °C for 3 h, then evaporated under hood and re-dissolved in 200 µL methanol for HPLC analysis after filtration using 0.2 μm sterile micro-filter. The supernatant of leaf extract was boiled for 30 min at 95 °C for preparation of hot extract.

### 4.5. Chromatographic Analysis

Standard solution (20 μL) or leaf extract was injected into the HPLC system LichroCart at a column temperature of 40 °C, using a RP-18 column (4 mm × 250 mm, 5 μm) (Merck, D-6100, Darmstadt, Germany). The mobile phase A consisted of water/containing 0.1% formic acid, and methanol was used as mobile phase B. The gradient conditions were as follows: Initially 25% A and 75% B, 0–39 min linear change to 55% A and 45% B, 39–40 min linear change to 25% A and 75% B. Gradient elution was achieved at a flow rate of 1 mL min^−1^. The accuracy of the HPLC method was evaluated six times by consecutive injection of the sample. Silymarin derivatives were detected at 288 nm.

### 4.6. Samples Preparation for Mass Spectrometric Analysis

The peaks eluting from HPLC separations of standard silymarin or leaf extract were collected with a fraction collector. Samples of interest were transferred into Eppendorf tubes pretreated with acetaldehyde. After evaporating the solvent in a vacuum centrifuge with cooling, the residues were dissolved in 50% *v*/*v* methanol. Samples of 150 μL volume were injected in the electrospray ionization mass spectrometry (ESI-MS) and MS and MS/MS analyses were performed with Bruker Daltonics, Esquire 3000 plus, Germany, using an ESI source with negative ion mode. The spray voltage was set to 4000 V. The temperature of the drying gas (N_2_) was kept at 180 °C. The ion trap of the ESQUIRE 3000 was used to trap ions of interest and keep them in a circular path within the trap. Subsequently, the radiofrequency was changed to fragment the collected ions. Initiation of the next scanning procedure allowed for determination of the fragment masses according to the recommendations of the manufacturer. We depict MS/MS-scans of four mother MS peaks for validation of apigenin-7-*O*-glucuronide (Figure 3), silybin A, silybin B, and isosilybin (Figure 4).

### 4.7. Preparation of Enzymes Extract

Fresh leaves (5 g) were homogenized in a Waring Blender containing 40 mL of 100 mM phosphate buffer, pH 7.5, containing 5 mM dithiothreitol, and 1.8 g polyclar AT for 15 s. The homogenate was then filtered through Pellon and the filtrate was centrifuged at 25,000× *g* for 15 min. The resulting supernatant represented the crude enzyme extract. The extraction process was carried out at 4 °C.

### 4.8. Enzymes Assays

Phenylalanine ammonia lyase (PAL, EC 4.3. 1.25). PAL activity was assayed by measuring the trans-cinnamic acid formation at 290 nm using a UV-1800 UV-Vis spectrophotometer and a standard curve of trans-cinnamic acid. The reaction mixture contained 100 mM Tris-HCl buffer (pH 8.8), 40 mM l-phenylalanine, and 200 µL of the enzyme preparation in a total volume of 1 mL. The reaction was carried out at 37 °C for 30 min and terminated by the addition of 50 µL of 4 M HCl. PAL activity was expressed as U/mg [60].

Chalcone synthase (*CHS, EC: 2.3.1.74*). CHS activity was assayed spectrophotometrically according to [61]. Enzyme was extracted by homogenizing the leaves (0.5 g) at 4 °C in 5 mL of 0.1 M borate buffer (pH 8.8) containing 1 mM 2-mercaptoethanol. The homogenates were treated with 0.1 g of Dowex l × 4 for 10 min and the cell debris were removed by centrifugation at 15,000 rpm for 10 min. Dowex l × 4 resin (0.2 g) was added to the supernatant and treated for another 20 min. The resin was then removed by centrifugation for 15 min at 15,000 rpm. The resultant supernatant was used for the CHS assay. For this, 100 μL of enzyme extract was mixed with 1.89 mL of 50 mM Tris-HCl buffer (pH 7.6) containing 10 mM KCN. The enzyme reaction was allowed to proceed at 30 °C for 1 min after adding 10 mg of chalcone in 10 μL ethylene glycol monomethyl ether. The activity was determined by measuring the absorbance at 370 nm.

Flavonoid 3′-monooxygenase (F3′H, EC 1.14.14.82). The assay mixture according to [62] contained 50 mM Bicine buffer (pH 8.5), 0.5 mM NADPH, 2 mM dithiothreitol, 200 µL of enzyme extract, and 0.5 mM kaempferol in a total volume of 200 µL. The reaction mixture was mixed, incubated at 30 °C for 30 min, and shaken in an open Eppendorf centrifuge tube. The reaction in which kaempferol was converted to quercetin was terminated by addition of 800 µL of a 2:1 (*v*/*v*) mixture of chloroform: methanol (containing 1% HCl) and mixing. The resulting biphasic solutions were then centrifuged at 15,000× *g* for 30 s. Produced quercetin was partitioned into the 400 µL methanolic upper phase. Specific activity of the hydrolase was defined as nmol of product formed per min per mg protein.

Shikimate O-hydroxycinnamoyl transferase (SHCT, EC 2.3.1.133). The assay was done according to [63]. The reaction mixture contained 50 mM potassium phosphate buffer (pH 6.5) 10 µmol hydroxycinnamoyl-CoA thiol ester; 2 µmol shikimic acid, and 50 µL enzyme extract in a final volume of 1 mL. The assay mix was incubated at 30 °C. The decrease in the absorbance of the reaction mixture was followed at 340 nm.

Cinnamoyl-CoA reductase (CCR, EC 1.2.1.44). The reaction mixture contained 0.1 mM NADPH, 50 µmol cinnamoyl-CoA ester, 100 mM phosphate buffer (pH 6.25), and 200 µL enzyme extract in a total volume of 1 mL. The oxidation of NADPH was determined by the decrease in absorbance at 366 nm. At this wavelength, the change in absorbance is the result of the decrease in the absorbance of cinnamoyl-CoA ester and NADPH and of the increase in absorbance of cinnamaldehyde. The reaction rate was calculated from the absorbance using the correcting factors according to [64].

Cinnamyl-alcohol dehydrogenase (CAD, EC 1.1.1.195). The enzymatic reduction of cinnamyl-aldehyde was determined by the change in absorbance at 340 nm according to [65]. The incubation mixture in a final volume of 1 mL consisted of 0.1 mM NADPH, 70 µmol cinnamyl-aldehyde, 100 mM phosphate buffer (pH 6.25), and 200 µL enzyme extract.

Caffeoyl-CoA O-methyltransferase (COMT, EC 2.1.1.104). The assay was carried out in a 96-well polystyrene microplate (Nunc, Roskilde, Denmark) that is compatible with a Tecan plate reader (Ultra 384, Salzburg, Austria). In each well, a total of 100 µL of reaction volume was conducted in assay buffer (50 mM Tris-HCl, pH 7.5) containing 500 µmol S-adenosyl-l-methionine (Adomet), 100 µmol caffeic acid, and 200 µL enzyme extract. The control consisted of extract inactivated by boiling. The reaction mixture was incubated at 30 °C for 15 min. The reaction was terminated by addition of 1% TCA followed by adding 20 µL of 0.4% (*w*/*v*) Gibbs’ reagent in ethanol for developing the dye complex. The absorbance was recorded at 625 nm for Gibbs’ reagent-ferulate complex [66].

### 4.9. Protein Determination

Determination of the soluble protein content was carried out according to [67]. One mL extract was mixed with 5 mL diluted Coomassie Brilliant blue (CBB) G-250. The mixture was kept in the dark for 1 min and the absorption was measured spectrophotometrically at 595 nm. The concentration of the protein was determined from standard curve using bovine serum albumin.

## 5. Conclusions

The current study demonstrates that metabolites produced by the undifferentiated tissue of *S. marianum* (callus) can be converted to pharmacologically important compounds using the enzymatic equipment of intact leaves or leaf extracts. The age of the leaves had a strong impact on their ability to catalyze the conversion of the primary products generated in the callus.

Based on the chromatographic separation and mass spectrometric identification, it is concluded that silymarin compounds can be produced from phenolic acids produced by the *S. marianum* callus. Efficient conversion occurred by incubating callus extract with medium-age leaf discs for 16 h. Alternatively, combining leaf extract obtained at pH 7.2 with callus extract for 3 h incubation at 37 °C also produced silymarin compounds. Interestingly, the leaves were unable to accumulate silymarin themselves, but leaf enzymes induced silymarin biosynthesis. Enzymes preparations from leaf of *S. marianum* catalyzed the oxidation of DCQ to flavonolignans. The reaction required a proper pH and revealed a pH optimum of about 7.4. Activities of enzymes of the phenylpropanoid and flavonoid pathway changed between young, medium, and old leaves in a manner supporting the above hypothesis. The here established bioconversion system opens up two perspectives, namely to use this system to identify additional enzymes required for the conversion of precursors to pharmacologically important drugs and to develop biotechnological approaches for bioorganic synthesis of silymarin compounds.

## Figures and Tables

**Figure 1 ijms-22-02149-f001:**
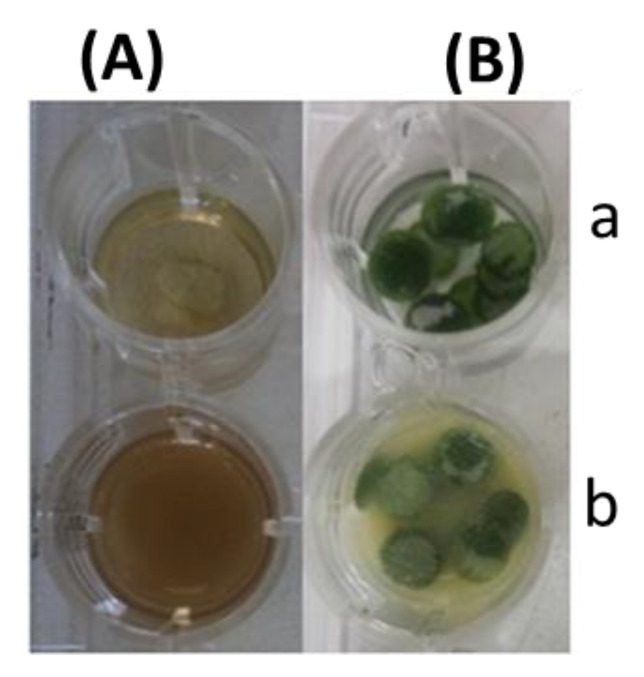
Bioconversion assay. (**A**) Reaction mixture of *S. marianum* leaf extract and callus extract (**B**) Leaf discs soaked with callus extract. (a) Control medium, (b) callus extract.

**Figure 2 ijms-22-02149-f002:**
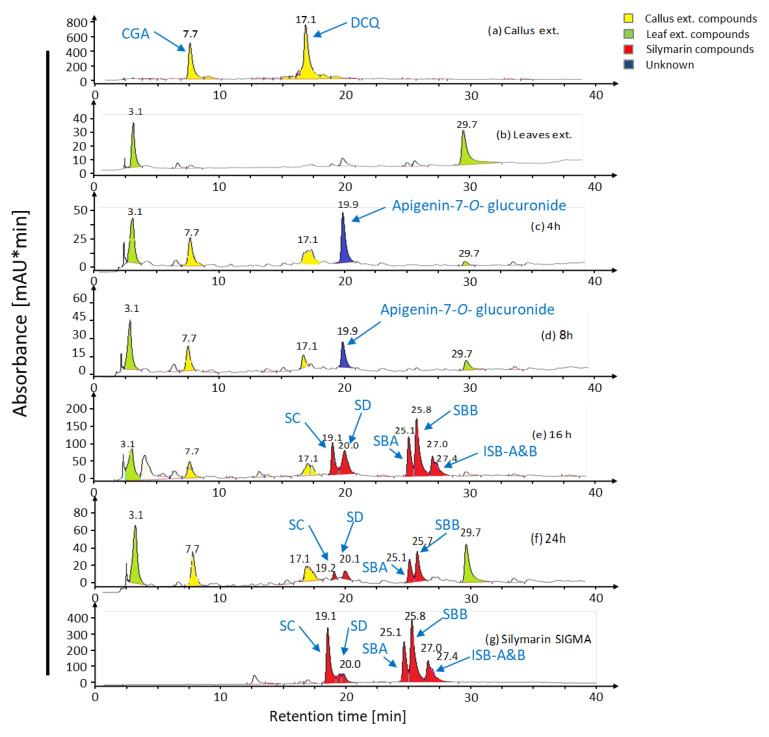
Chromatographic analysis of the products obtained from bioconversion. HPLC separations show the successive bioconversion of (**a**) callus compounds to silymarin derivatives using *S. marianum* leaf discs (**b**) and callus extract for 4 h (**c**), 8 h (**d**), 16 h (**e**), and 24 h (**f**). The conversion profiles are compared with standard silymarin SIGMA (**g**). Data obtained derive from six independent experiments.

**Figure 3 ijms-22-02149-f003:**
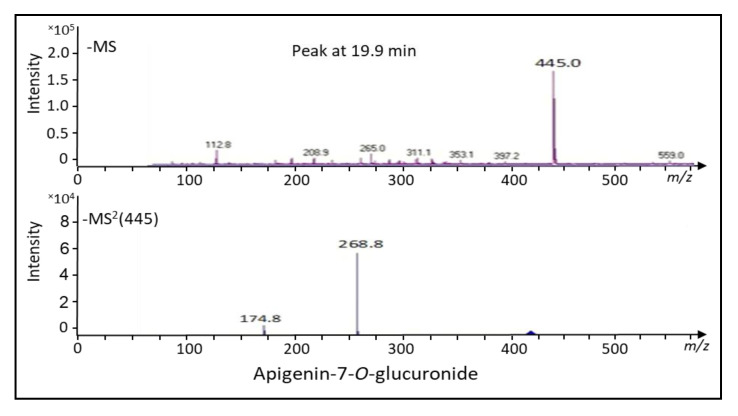
Electrospray ionization mass spectrometry (ESI-MS) and MS/MS fragmentation pattern of the peak released at 19.9 min in the HPLC chromatogram. The peak obtained from Figure 2c was subjected to analysis in the negative mode of the ion for peak at 19.9 min via ESI-MS and MS/MS technique. The obtained *m*/*z* values reflect the molecular mass of apigenin-7-*O*-glucuronide (*m*/*z* = 445.0).

**Figure 4 ijms-22-02149-f004:**
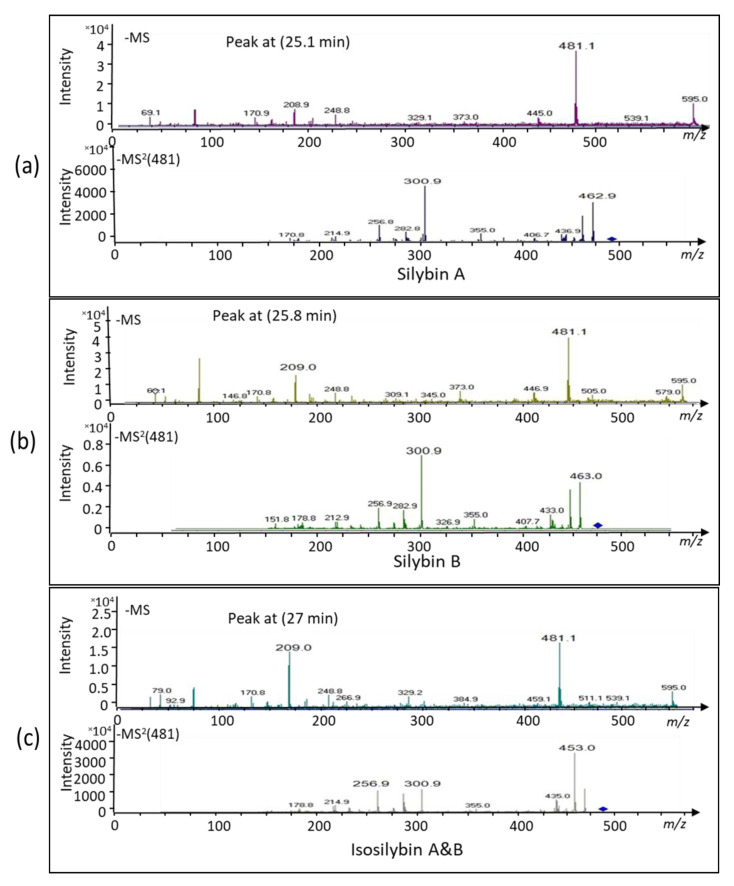
ESI-MS and MS/MS fragmentation pattern of the peaks eluted at 25.1, 25.8, and 27 min in the HPLC chromatogram. (**a**) ESI-MS and MS/MS in the negative mode of the ion peak at 25.1 min. The obtained *m*/*z* values of 481.1 corresponded to the molecular mass of both silybin A and B. The assignment to silybin A followed the literature using the HPLC retention pattern where the first peak is A and the second B [7]. Also, the patterns obtained by MS^2^ fragmentation of the *m*/*z*-peak at 481.1 were highly similar for both peak at 25.1 and 25.8 (**b**) confirming the correctness of assignment to silybin A and B. The MS^2^ peak at 300.9 matched the main peak reported in [7], but also the peaks at 256.9 and 282.9, further confirming the presence of silybin A and B. (**c**) The obtained *m*/*z* values for the peak at 27 min matched the molecular mass of isosilybin A and B.

**Figure 5 ijms-22-02149-f005:**
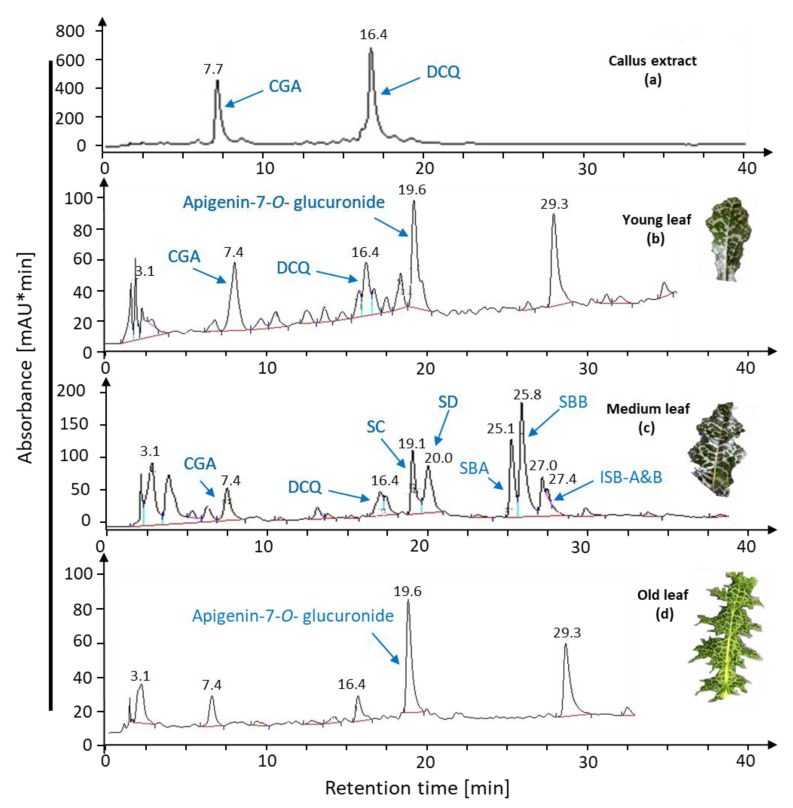
HPLC chromatograms of bioconversion assays with young, medium, and old leaves. The callus extract shown in (**a**) was exposed to leaf discs of young (**b**), medium (**c**), and old (**d**). Leaves from greenhouse-grown *S. marianum*. Data are representative for six independent experiments.

**Figure 6 ijms-22-02149-f006:**
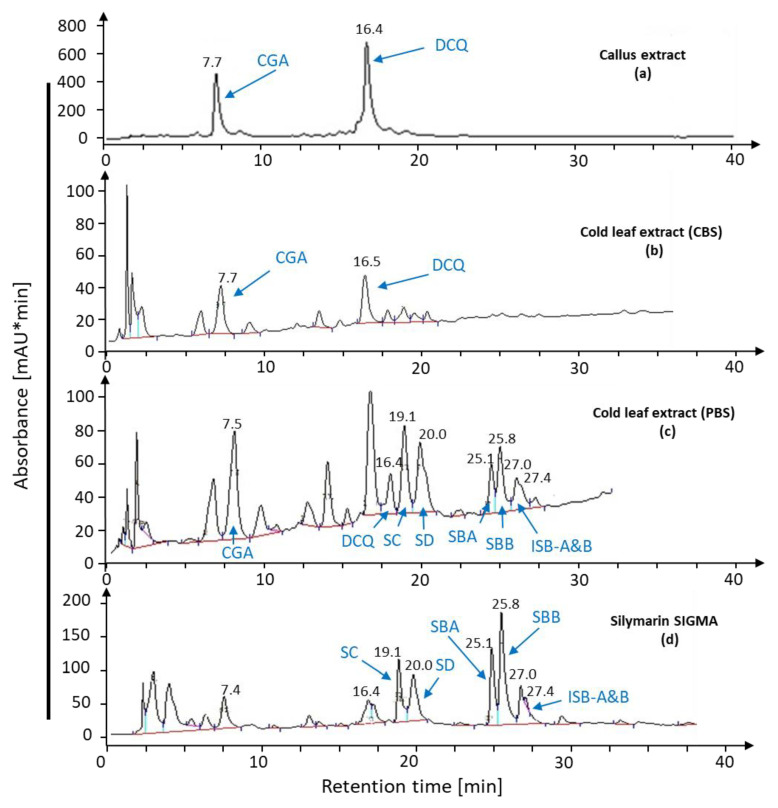
Bioconversion with cold leaf extract. Leaf extract obtained with either phosphate-buffered solution at pH 7.2 or citric acid buffer at pH 4.6 was incubated with callus extract and analyzed for silymarin biosynthesis after incubation for 3 h.

**Figure 7 ijms-22-02149-f007:**
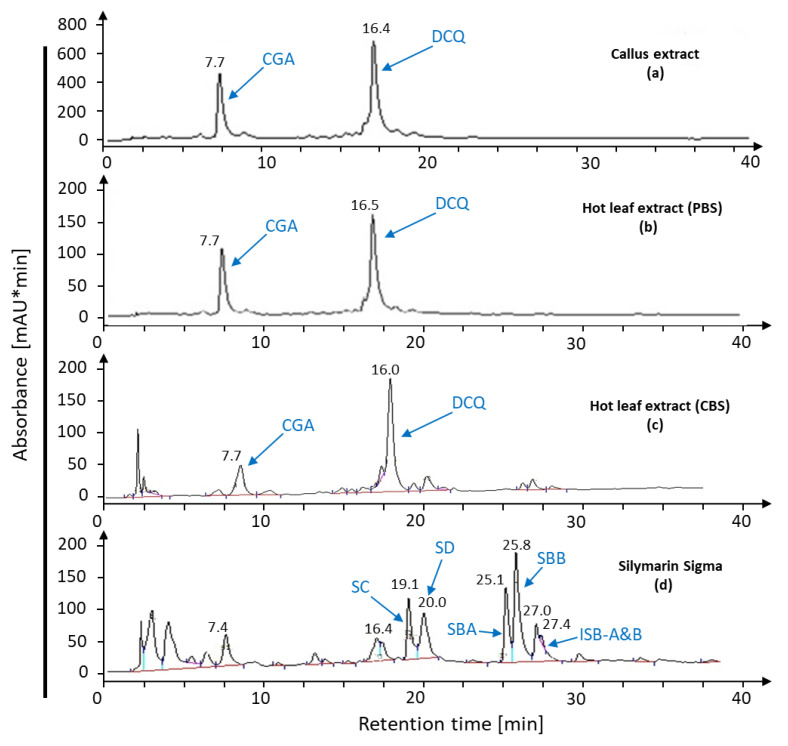
Bioconversion assay with hot leaf extract. Callus extract (**a**) was incubated with hot leaf extract prepared with citric acid buffer, pH = 4.6 (**c**), or phosphate buffer, pH 7.2 (**b**), on silymarin biosynthesis after incubation for 3 h. Profiles were compared with silymarin standard (**d**).

**Figure 8 ijms-22-02149-f008:**
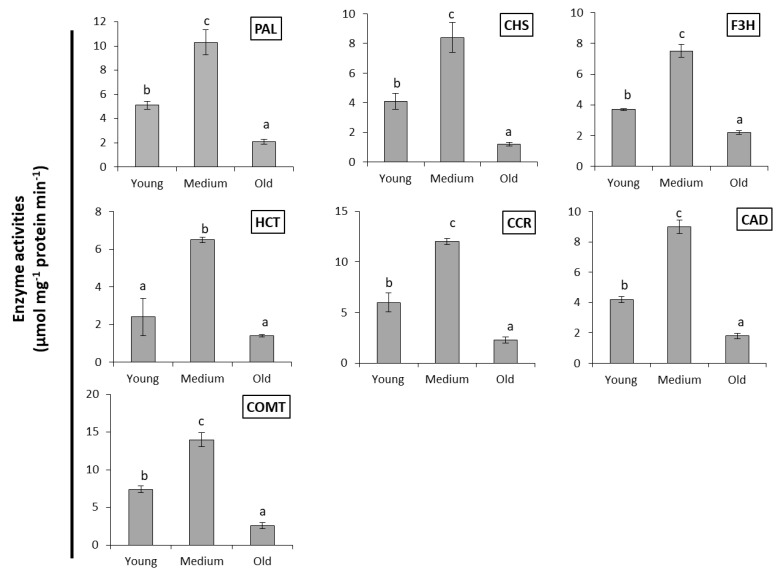
Enzyme activities (µmol mg^−1^ protein min^−1^) of phenylalanine ammonia lyase (PAL), chalcone synthase (CHS), flavonoid 3′-monooxygenase (F3′H), shikimate *O*-hydroxycinnamoyl transferase (HCT), cinnamoyl-CoA reductase (CCR), cinnamyl-alcohol dehydrogenase (CAD), and caffeoyl-CoA *O*-methyltransferase (COMT) in *S. marianum* leaves of different developmental stages. Data represent the mean ± SD (*n* = 3). Different letters indicate significant difference (*p* < 0.01) assessed by Tukey–Kramer’s test.

**Figure 9 ijms-22-02149-f009:**
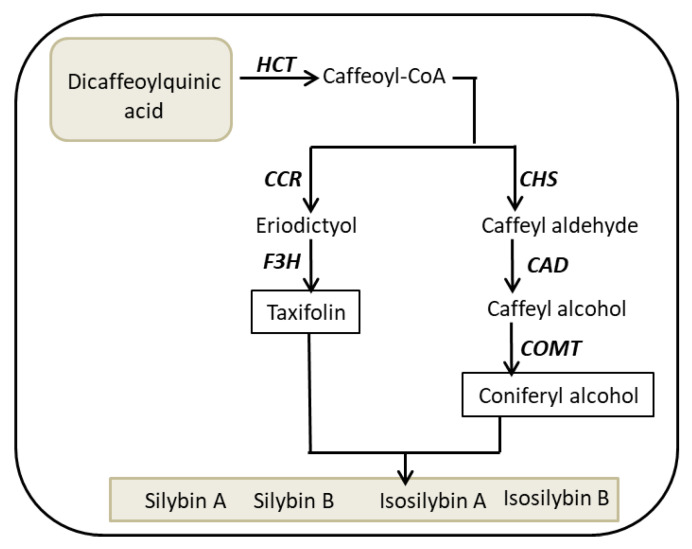
Proposed metabolic pathway for silybin biosynthesis. Callus extract contains dicaffeoylquinic acid (*S. marianum* callus components), which is converted to caffeoyl-CoA that further metabolizes to either taxifolin or coniferyl alcohol. Both taxifolin and coniferyl alcohol are combined for synthesis of silymarin compounds.

**Table 1 ijms-22-02149-t001:** Production of silymarin (mg·g callus DW^−1^) from callus extract by using *S. marianum* leaf discs or leaf extract during incubation for 16 h and 3 h, respectively. Chromatograms obtained by HPLC separations for quantification of silymarin fractions, silychristin (SC), silydianin (SD), silybin A (SBA), silybin B (SBB), and isosilybin A and B (ISB-AB). The values are mean ± standard deviation (SD).

Content (mg·g DW^−1^)	SC	SD	SBA	SBB	ISB-AB		Total
Leaf discs	1.3 ± 0.5	2.2 ± 0.8	1.6 ± 0.2	1.6 ± 0.2		1.5 ± 0.03	8.2
Leaf extract	1.09 ± 0.5	1.13 ± 0.5	0.82 ± 0.09	0.72 ± 0.05		0.86 ± 0.4	4.6

## Data Availability

Not applicable.

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
