# Peer review of "Bioconversion of Callus-Produced Precursors to Silymarin Derivatives in Silybum marianum Leaves for the Production of Bioactive Compounds"

_ijms, 2021, doi:10.3390/ijms22042149_

Round 1
Reviewer 1 Report
The manuscript "Bioconversion of callus-produced precursors to silymarin derivatives in Silybum marianum leaves for the production of bioactive compounds" by D. Gad et al. is devoted to the possibility of modifying the metabolites of the milk thistle callus using the enzymes of the leaves of the plant. The manuscript is well and logically organized. However, the authors must make a number of changes in order to make the manuscript accepted:
- The retention time of the compounds in Figures 5 (b,d), 6 (b,c) and 7 (c) does not correspond to the given scale of chromatograms.
- Authors should carefully check the correctness of the list of references: everywhere use the abbreviation of the journal (Ref. 8,12,15, etc.) and punctuation when abbreviating the name of the journal (Ref. 2,21, etc.); only the first word in the title of the article should be capitalized (Ref. 16, 25,35, etc.).
In addition, the reviewer has a number of questions about the manuscript:
- I have not seen experiments on biotransformation dynamics using leaf extract. How do the authors explain the difference in the yields of biotransformation products obtained using leaf discs and leaf extract?
- Was it necessary to observe aseptic conditions during the biotransformation process?
- What criteria did the authors use to classify leaves into young, medium and old?
Author Response
Response to Comments from Reviewer 1
Notes: General changes suggested:
- The retention time of the compounds in Figures 5 (b,d), 6 (b,c) and 7 (c) does not correspond to the given scale of chromatograms.
Response: Done. Please check figure 5, 6 and 7.
- Authors should carefully check the correctness of the list of references: everywhere use the abbreviation of the journal (Ref. 8,12,15, etc.) and punctuation when abbreviating the name of the journal (Ref. 2,21, etc.); only the first word in the title of the article should be capitalized (Ref. 16, 25,35, etc.).
Response: Done. Please check references.
- I have not seen experiments on biotransformation dynamics using leaf extract. How do the authors explain the difference in the yields of biotransformation products obtained using leaf discs and leaf extract?
Response: We performed the bioconversion of callus compounds (DCQ) to silymarin derivatives by two methods: one of them used leaf discs (Fig. 1B) and the second method used leaf extract prepared in phosphate or citric acid buffer together with and hot and cold extract as shown in (Fig. 1A, 6 and 7). The details are illustrated on page 6 line 147 and the methods are described on page 12 line 314. The compounds produced in both approaches were estimated quantitatively by HPLC as shown in Table 1.
- Was it necessary to observe aseptic conditions during the biotransformation process?
Response: No, it was not necessary to perform the biotransformation under aseptic conditions.
- What criteria did the authors use to classify leaves into young, medium and old?
Response: The authors classify the age of the leaf in to young, medium and old according to morphology and size of the leaf. Young leaf were foliage leaf (not cotyledons) of small size and an area of about 11x4.5 cm(2). The leaves still expanded and the margin spines were still soft and not sharp. The leaf colour was pale green. Medium leaf were fully expanded mature blade with short petiole and sharp spiny margin with green colour and white veins. The leaf areas were about 17x7 cm(2). Old leaves had leaf blades that were lobed. They had dark green colour with very hard spines and long petioles.
According to these criteria, it was easy to distinguish between medium and old leaves while young leaves also required careful investigation of the positioning along the shoot.
Reviewer 2 Report
Authors should: 1. correct a series of small mistakes reported highlighted in yellow in theattached file
2. report references as requested by the journal 3. it would be useful to specify well what is meant by silymarin, silybinin
and silybins. Silymarin is a mixture of about thirty substances of which
the molecules analyzed in the article are perhaps only the most abundant.

Author Response
Response to Comments from Reviewer 2
Reviewer #2:
- correct a series of small mistakes reported highlighted in yellow in the attached file.
Response: Done, with appreciation.
- report references as requested by the journal.
Response: Done.
- it would be useful to specify well what is meant by silymarin, silybinin and silybins. Silymarin is a mixture of about thirty substances of which the molecules analyzed in the article are perhaps only the most abundant.
Response: We changed to the word “sum” indicating that the given number corresponds to the sum of the individual data in the table.
Round 2
Reviewer 2 Report
Some errors in the layout of the references. Still publishable work
Author Response
Response to Comments from Reviewer 2
We revised the manuscript according to the suggestions of the reviewer. We are grateful to the reviewers for the suggestions that have surely strengthen our manuscript.
Reviewer #2:
- Some errors in the layout of the references. Still publishable work
Response: Done,